# Inorganic Carbon Assimilation and Electrosynthesis of Platform Chemicals in Bioelectrochemical Systems (BESs) Inoculated with *Clostridium saccharoperbutylacetonicum* N1-H4

**DOI:** 10.3390/microorganisms11030735

**Published:** 2023-03-13

**Authors:** Rosa Anna Nastro, Anna Salvian, Chandrasekhar Kuppam, Vincenzo Pasquale, Andrea Pietrelli, Claudio Avignone Rossa

**Affiliations:** 1Department of Science and Technology, University of Naples “Parthenope”, 80133 Naples, Italy; 2Laboratory of Systems Microbiology, Department of Microbial Sciences, University of Surrey, Guildford GU2 7XH, UK; 3Department of Biotechnology, Vignan’s Foundation for Science, Technology and Research, Vadlamudi, Guntur 522213, Andhra Pradesh, India; 4Laboratoire Ampere CNRS UMR 5005, Département Génie Electrique et des Procédés Université de Lyon, F-69621 Villeurbanne, France

**Keywords:** CO_2_ capture, *Clostridium saccharoperbutylacetonicum* NT-1, electrosynthesis, 3-D-hydroxybutyrate, PCA, *Shewanella oneidensis* MR1, *Pseudomonas aeruginosa* PA1430/CO1

## Abstract

The need for greener processes to satisfy the demand of platform chemicals together with the possibility of reusing CO_2_ from human activities has recently encouraged research on the set-up, optimization, and development of bioelectrochemical systems (BESs) for the electrosynthesis of organic compounds from inorganic carbon (CO_2_, HCO_3_^−^). In the present study, we tested the ability of *Clostridium saccharoperbutylacetonicum* N1-4 (DSMZ 14923) to produce acetate and D-3-hydroxybutyrate from inorganic carbon present in a CO_2_:N_2_ gas mix. At the same time, we tested the ability of a *Shewanella oneidensis* MR1 and *Pseudomonas aeruginosa* PA1430/CO1 consortium to provide reducing power to sustain carbon assimilation at the cathode. We tested the performance of three different systems with the same layouts, inocula, and media, but with the application of 1.5 V external voltage, of a 1000 Ω external load, and without any connection between the electrodes or external devices (open circuit voltage, OCV). We compared both CO_2_ assimilation rate and production of metabolites (formate, acetate 3-D-hydroxybutyrate) in our BESs with the values obtained in non-electrogenic control cultures and estimated the energy used by our BESs to assimilate 1 mol of CO_2_. Our results showed that *C. saccharoperbutylacetonicum* NT-1 achieved the maximum CO_2_ assimilation (95.5%) when the microbial fuel cells (MFCs) were connected to the 1000 Ω external resistor, with the *Shewanella*/*Pseudomonas* consortium as the only source of electrons. Furthermore, we detected a shift in the metabolism of *C. saccharoperbutylacetonicum* NT-1 because of its prolonged activity in BESs. Our results open new perspectives for the utilization of BESs in carbon capture and electrosynthesis of platform chemicals.

## 1. Introduction

### CO_2_ Capture by Microorganisms

The ability of both autotrophic and heterotrophic microorganisms to use carbon in its inorganic form has been known and studied for quite a long time [1,2,3,4]. The evidence of its spread among microorganisms is becoming stronger with the discovery of new metabolic pathways. It has been estimated that CO_2_ utilization by microorganisms dates back to the earliest life forms on the Earth, with the reductive tricarboxylic acid cycle (rTCA) and the Wood–Ljungdahl pathway (WL) recognized as the most ancient carbon fixation pathways [5]. Besides the rTCA and WL pathways and the well-known Calvin–Benson–Bassham (CBB), other processes involved in CO_2_ metabolism have been recently discovered [6]. Fatty alcohols and acids biosynthesis in *Pseudomonas aeruginosa* and the reversed-pyruvate ferredoxin oxidoreductase (rPFOR)/pyruvate-formate-lyase-dependent (Pfl) carbon assimilation in *Clostridium beijerincki* are two examples of newly described mechanisms allowing bacteria other than chemolithotrophic to use inorganic carbon [7,8,9]. It has been demonstrated that the assimilation of CO_2_ can give bacteria some advantages, such as dissipating the excess of reducing equivalents produced by metabolic processes or accumulating organic molecules as energy and carbon storage [6]. Among the microorganisms able to use inorganic sources of carbon, clostridia have been extensively investigated over the years due to their metabolic potential for industrial applications, with significant results [10,11]. Different strategies for CO_2_ capture, conversion, improvement, and control have been developed, involving genetic engineering, synthetic biology, metabolic engineering, and more recently, microbial electrosynthesis (MES) [11,12,13], which operates at the nexus of microbiology and electrochemistry to reduce CO_2_ to value-added products [12,13]. MES is especially performed in microbial electrochemical technology (MET) which has shown its potential in different applications, including CO_2_ capture and assimilation [14,15]. Various studies have been reported confirming the potential applications of MET for the conversion of CO_2_ into platform chemicals such as CH_4_, organic acids (formic and oxalic acids, among others), ethanol, and volatile fatty acids (VFAs), and strategies for scaling up such systems are being developed as well [15,16,17,18,19,20]. The performance of MET can vary according to the operational conditions, the microbial species, and the substrates used. For example, organic substrates in the catholyte might reduce or even prevent CO_2_ assimilation, fostering the establishment of a heterothrophic instead of an autothrophic or mixothrophic metabolism. Therefore, the set-up of MET devices requires adequate preliminary studies [17,21,22].

In typical bioelectrochemical systems (BESs) for electrosynthesis, bacteria able to assimilate CO_2_ are inoculated in the catholyte of a two-chamber (or even single chamber) microbial electrochemical cell (MEC), provided with ions exchanging membrane. An external potential, provided by a potentiostat or a battery, is used to “drive” the metabolic processes at the cathodes [12,15,16,17,21,23,24]. The main reason why an external source of energy is required to realize the conversion of inorganic carbon into valuable organic molecules lies in the need of reducing equivalents (electrons) to sustain the whole process and drive the microbial metabolism towards the synthesis of target compounds. Nevertheless, it is possible to set up a system in which reducing equivalents, which in natural environment are mostly provided by H_2_, can derive from the degradation of reduced substrates by the activity of exoelectrogenic bacteria growing at the anode [14,15]. Similar processes occur in anaerobic/anoxic natural environments and are at the basis of an efficient direct/indirect interspecies electron transfer (IET) and the establishment of a cooperative catabolism [25]. 

Within this framework, we assessed the efficiency of a fermentative microorganism such as *Clostridium saccharoperbutylacetonicum* NT-1, used in industrial processes for the production of bio-butanol and acetone [26], to capture CO_2_ from a gas mix and in the presence and absence of an external source of applied electrochemical potential. We chose to express the amount of CO_2_ captured from a gas mix as a percentage as it allows direct comparison between BESs and other already available carbon capture technologies, based on both abiotic and biotic processes [27].

We used a consortium *Pseudomonas aeruginosa* PA1430/CO1/*Shewanella oneidensis* MR_1 as bioanode and investigated their ability to sustain CO_2_ assimilation from a gas mix in *C. saccharoperbutylacetonicum* NT-1 biocathodes, previously established in controlled growth conditions. We also investigated potential modifications in *C. saccharoperbutylacetonicum* NT-1 metabolism in terms of utilization of carbon sources after a prolonged utilization in BESs. To the best of our knowledge, this is the first time that an approach envisaging the utilization of both MFCs, MECs, and pre-constituted bioelectrodes has been used.

## 2. Materials and Methods

### 2.1. Microorganisms and Culture Media

*Clostridium saccharoperbutylacetonicum* NT-1 was obtained from the DSMZ collection (DSMZ 14923) and cultured according to the supplier’s instructions. We inoculated the strain in double replicas using clostridium reinforced medium (CRM, Oxoid, Basingstoke, UK) and incubated under anaerobic atmosphere (2% CO_2_, 98% N_2_) at 30 °C for 48 h. Then, 100 μL of the cultures was spread on 9 cm CRM plates, supplemented with 15% *w*/*v* bacteriological agar (Oxoid, Basingstoke, UK). Isolated colonies were transferred to a modified tryptone-soya yeast extract medium (TSYEM, 1/4 strength, without sucrose) composed of tryptone 2.5 g/L, yeast extract 2.5 g/L; (NH_4_)_2_SO_4_ 0.5 g/L; FeSO_4_ 0.025 g/L. pH was adjusted to 6.5 ± 0.2 with 0.1M 2-(N-morpholino)ethanesulfuronic acid (MES). After 48 h of incubation at 30 °C in anaerobiosis, 100 μL of microbial suspension was inoculated in 9.9 mL of a further modified TSYEM, where tryptone was replaced with NaHCO_3_ (0.5% *w*/*v*), final pH 6.8 ± 0.2 The catholyte for CO_2_ capture from a gaseous mix was composed of yeast extract (YE) 2.5 g/L; (NH_4_)_2_SO_4_ 0.5 g/L; FeSO_4_ 0.025 g/L, with 6.8 ± 0.2 as final pH. YE was added as a necessary component for clostridia growth and metabolic functions, supplying micronutrients and growth factors [22,24,28]. The bioanode strains, *Shewanella oneidensis* MR1 (ATCC 700550) and *Pseudomonas aeruginosa* PA1430/CO1 (DSM 19882), were cultured in nutrient broth (NA, Oxoid, Basingstoke, UK) according to the supplier’s instructions and incubated at 30 °C for 24 h. Both strains were subcultured in tryptic soy agar (TSA, Oxoid, Basingstoke, UK) plates, pH 7.2 ± 0.2, and maintained for use in later experiments. Tryptic soy broth (TSB) and M9 medium [29] containing glycerol (0.4%) as sole source of carbon were used to culture both strains and prepare the bioanodes for our BESs. The final composition (per 1 liter) was MgSO_4_ 1 mM, CaCl_2_ 0.3 mM, biotin 1 µg, thiamin 1 µg, 10 mL of trace elements solution (100X), 100 mL of salt solution (10X), and glycerol 0.4%. Final pH was 7.2 ± 0.2. The same M9 medium was used as anolyte in both MFCs and MECs [30,31]. The co-cultures were incubated at 30°C for 24 h, in an orbital shaker (150 rpm). At the end of the incubation, 2 mL of each co-culture was added to 18 mL of fresh medium and incubated for further 48 h to be inoculated in the bioanodes. The presence and viability of both strains was assessed preparing serial solutions in 0.9% NaCl and following culture on pseudomonas isolation agar (PSI) and TSA, all purchased at Oxoid©. 

### 2.2. Bioanode and Biocathode Protocol

We assembled single chamber, air-cathode MFCs by using 100 mL Duran bottles with two carbon cloth anodes (2.5 cm × 3.5 cm and 1 mm thickness) (NCBE, University of Reading, Reading, UK) connected to each other through a 100 Ohm resistor by means of a 0.3 mm Ni/Cr 80/20 wire (RS Components, Corby, UK). The systems were completed by a graphite rod (0.5 cm diameter, 7 cm length) acting as a “snorkel”, i.e., an electrode able to increase the electrochemical reactions rate [32]. The MFCs final configuration was open circuit voltage (OCV) [33]. Biofilm formation was carried out under shaking conditions (150 rpm/min).

The preparation of the *C. saccharoperbutylacetonicum* NT-1 biofilms was as follows: 1 mL inoculum of starting microbial cultures (OD_600_ = 0.5) was added to 99 mL of CRM in 100 mL Duran bottles in the presence and absence (for no-BESs control cultures) of electrodes, followed by incubation under anaerobic conditions at 30 °C for 48 h. We transferred the cathodes every 24 h in two steps: first from CRM to TSYEM, and then to TSYEM with 0.5% NaHCO_3,_ before assembling our BESs and sparging the gas mix. Along with the bioelectrodes, we prepared control cultures by transferring 10 mL of *Clostridium saccharoperbutylacetonicum* NT-1 initial culture in CRM in 90 mL of TSYEM with complete formula. After a further incubation, 10 mL of microbial suspension in TSYEM was inoculated in 90 mL TSYEM with NaHCO_3_ following the procedure described above. Finally, the culture was transferred to mineral medium with the gas mix as the sole carbon source. For the anodic co-cultures, we inoculated 1 mL of 1 OD_600_ liquid cultures of each strain to 8 mL of tryptone soy broth (TSB). We subsequently transferred 1 mL of a fresh co-culture (0.5 OD_600_) of *S. oneidensis* M41 and *P. aeruginosa PA1430/CO1* to 99 mL of TSB in 100 mL Duran bottles. As previously performed for *C. saccharoperbutylacetonicum* NT-1, the cultures were incubated at 150 rpm/min for 48 h at 30 °C, and then we replaced the TSB with M9 medium. We incubated the M9 culture at 30 °C for 72 h, with medium changed every 24 h. We measured electrode potentials vs. Ag/AgCl along with the change of the medium by posing E_pot_ = ΔV − E_Ag/AgCl_, where E_pot_ is the potential of the electrode in the MFCs. We also measured the ORP and pH of control cultures and MFCs medium every 24 h [34]. Further details about the methodological approach we used to set up the bioanodes and biocathodes are described in Appendix A.

### 2.3. Set-Up of Two Chamber-MFCs for CO_2_ Capture

We used two-chamber MFCs and MECs with an overall volume of 250 mL to perform further testing activity and electrochemical analyses. We prepared a double-folded carbon cloth (58.5 cm^2^ surface area), with a Ni/Cr wire (80/20) used as the electron collector. Given the higher volume of catholyte in the 250 mL cells in comparison to the 20 mL systems, we set up composite cathodes made of carbon cloth (40 cm^2^) wrapped with a Ni/Cr wire, used also as the electron collector, to a graphite rod of 12.5 × 0.5 cm to improve cathode volume and the overall conductivity of the electrode [35]. We prepared both anode and cathode biofilms following the same procedure previously described for the 20 mL MFCs and MECs, i.e., by culturing electroactive biofilms in Duran bottles and gradually replacing the carbon source to HCO_3_^−^ for *C. saccharoperbutylacetonicum* NT-1 and glycerol for the anodic consortium. A cation exchange membrane (Fumatec FKE-50) mediated the ionic flux between the anode and cathode chambers (Figure 1). We applied a double layer of polyester tissue (J-cloth, 0.2 mm thickness) to both sides of the cations, exchanging membranes in all MFCs to reduce biofouling occurrence. In all MECs and MFCs, we placed the electrodes colonized by *Shewanella oneidensis* M1/*Pseudomonas aeruginosa* PA1430/CO1 and electrodes previously colonized by *C. saccharoperbutylacetonicum* NT-1 cultures, respectively, at the anode and the cathode chambers. As previously stated, M9 medium containing 0.4% glycerol and TSYEM added with NaHCO_3_ (0.5% final concentration) were used as anolyte and catholyte. After the first 24 h of operation, we replaced the HCO_3_^−^ in the TSYEM medium with CO_2_ by bubbling the CO_2_/N_2_ gas mix (2% CO_2_) in the cathode compartment. The amount of CO_2_ released from the MFCs and cultures’ headspaces was measured by an Electrolab CO_2_/O_2_ Off-Gas Analyser (Electrolab Biotech Ltd., Tewkesbury, UK). A graphical description of our experimental set-up is shown in Figure 1.

The BESs were set up as follows: MFCs connected to a 1000 Ω external load (MFC_1000 Ω) and MECs connected to a 1.5 V external battery so that the potential applied at the cathode was −750 mV vs. SHE (−955 mV vs. Ag/AgCl) and +750 mV (+545 mV vs. Ag/AgCl) at the anode. Nevertheless, a direct measurement revealed a drop of electrode potential, with a final value of −820 mV at the cathode and +550 mV vs. Ag/AgCl [17,24].

All the assays were performed in situ by considering the cathode as the working electrode and the Pt electrode as the counter electrode against the Ag/AgCl (3.0 M KCl) reference electrode. All the potentials mentioned further in the manuscript are vs. the Ag/AgCl reference electrode, unless otherwise stated. MFCs kept in OCV were used as control systems to evaluate the CO_2_ capture under open circuit conditions [36]. Other control systems such as MEC_blank and MFC_blank were used to evaluate the influence of just chemical–physical processes on CO_2_ sequestration from the gas mix. Control cultures served to assess the efficiency of *C. saccharoperbutylacetonicum* NT-1 to assimilate CO_2_ in the culture conditions, but in absence of electrodes and energy inputs. Finally, the sterile medium was used to evaluate CO_2_ reduction in the gas mix in consequence of the sole dissolution into the medium. All MECs, MFCs, and controls were set up in double replicas for three cycles of experimental activity, performed at 20 °C, with each cycle lasting about three weeks, considering also the preparation of pre-colonized bioanodes and biocathodes and the performance of chemical analyses On the whole, BESs and control cultures operated for five days before being disassembled.

### 2.4. Metabolic Shift in C. saccharoperbutylacetonicum NT-1

With the exception of the first experimental session, in each cycle of assays, we used acclimatized isolates from previously tested BESs: microorganisms used in the 3rd assay came from the systems used in the 2nd experiment and these last ones were inoculated with bacteria from the 1st round of tests. In order to investigate the metabolic patterns of the original strain and the one isolated from the cathode of a MFC-1000 Ω, we prepared fresh cultures of *C. saccharoperbutylacetonicum* NT-1 from the lyophilized culture and from strains previously used for several weeks at the cathode of an MFC connected to a 1000 Ω resistor in CO_2_ capture experiments. We used well-separated colonies previously cultured on CRM medium + bacteriological agar (15%) to prepare microbial suspensions in 0.9% NaCl solution (0.2 MacFarland final concentration). Then, we inoculated aliquots of 200 μ of 0.2 MacFarland suspension in ECOLOG 96-multiwell plates (BIOLOG, Hayward, CA, USA), sealed in a plastic bag with wet paper sheets, and anaerobically incubated at 30 °C. We measured the absorbance of the ECOLOG plates after 24 and 144 h at 590 nm with a Clariostar Multimode Microplate Reader (ThermoFisher, Waltham, MA, USA) and calculated the average well color development (AWCD) to quantify the ability of the microorganisms to use different organic substrates as a source of carbon and energy [37]. The 31 substrates in the BIOLOG ECO microplates were classified into six categories, namely carboxylic acids, carbohydrates, amino acids, polymers, phenolic compounds, and amines/amides. Further details are available in Appendix A.

### 2.5. CO_2_ Assimilation Efficiency

MFCs, MECs, and controls operated the first 48 h with 0.5% HCO_3_^−^ as a source of carbon before all media were replaced with mineral salts media and the gas mix was sparged at the bottom of the catholytes and controls by sterile needles inserted in an inflow hole. All BESs and control cultures were provided with a second hole to allow the exit of gas mix and its collection by a sterile tubing (3 mm diameter) for direct CO_2_ measurements with the Electrolab CO_2_ m. CO_2_ concentration in the cathode compartment headspace was measured after initial overnight sparging of the gas mix in BESs, control cultures, and abiotic blanks, under a pressure of 1.2 ± 0.2 atm. We subsequently replaced the medium with fresh TSYEM without carbon source and bubbled the gas mix for 15 min before making a first measurement. CO_2_ percentage in the headspace of the BESs, the control cultures, and blanks was then recorded after 6 h of further gas sparging after verifying the absence of significant fluctuation of CO_2_ in all samples. We then took 10 mL of catholyte and culture media from each replica for the chemical analyses. We calculated the CO_2_ capture rate, the carbon recovery (CR) defined as the percentage of inorganic carbon converted into target metabolites [28], and the energy required by both systems. The calculations are presented in the Appendix A.

### 2.6. Energy Balance and CO_2_ Assimilation

While injecting CO_2_ in the catholyte, we monitored the voltage of both MFCs_1000 Ω and MECs over time by connecting the systems to the fuel cell test equipment (ARBIN Instruments, College Station, TX, USA). Then, we calculated the current, power, and energy produced by the MFCs over time by applying Ohm’s law. The energy provided by electric systems can be expressed as Wh = V × A × h^−1^, with 1 Wh = 3600 J. We calculated the total energy produced by MFC_1000 Ω over time and expressed it as J/m^2^, where m^2^ refers to the cathode-projected surface.

We estimated the energy (*e*) needed to assimilate 1 mol of gas, according to Equation (1):(1)$$e=\frac{nCO_{2ass}}{Ei}$$
where *nCO_2ass_* are the moles captured by MFCs_1000 Ω and *Ei* is the energy expressed in J/m^2^ and produced by the system along the CO_2_ sparging. Although it is possible to express the energy spent in the electrosynthetic process as a function of Wh per mole of a specific organic by-product [28], in this paper, the energy values are expressed as J/m^2^ mol CO_2_. We applied the same approach to calculate the amount of energy provided by the external battery in MECs during the CO_2_ assimilation test. In this case, we estimated the amount of energy provided by the external battery according to Kirchhoff’s law [38], Equation (2):*I_b_* = (*V_b_* − *OCV_MEC_*)/*R_MEC_*(2)
where *I_b_* is the current provided by the battery *b* and *OCV_MFC_* is the open circuit voltage of MEC after being disconnected from the battery and achieving a constant voltage. *R_MEC_* is the internal resistance of MEC. The overall power provided to the cells was calculated as follows:(3)$$P_{b}=V_{b}\;\times \;I_{b}=V_{b}\;\times \;[(V_{b}-OCV_{MEC})/R_{\mathrm{MEC}}]$$
where *P_b_* and *V_b_* are, respectively, the electric power and the voltage provided by the battery, *I_b_* is the current intensity flowing through the system, *OCV_MEC_* and *R_MEC_* are the stable voltage value that MECs achieve once they are disconnected by the external battery, and *R_MEC_* is the internal resistance of MEC. Starting from *P_b_* values, we calculated the overall energy amount provided to the systems during the CO_2_ assimilation experiment, as previously described.

### 2.7. Electrochemical Analyses

At the end of the CO_2_ assimilation tests, we carried out polarization experiments to MFC_1000 Ω and MFC_OCV. In order to perform the polarization experiments, we disconnected the MFC_1000 Ω from the external resistor for two hours, or until it achieved a stable OCV. All polarization experiments were carried out using the Arbin fuel cell test equipment, over a range of resistances from 120 kΩ to 100 Ω (18 steps), with a 5 min time step. We performed cyclic voltammetry (CV) over the range −1 V to +1 V on the cathode colonized by *C. saccharoperbutylacetonicum*-NT1 prior to its utilization in BESs, MFC_1000 Ω, MEC, and MFC_OCV. Tests were carried out at pH of 6.8 ± 0.2 using a PALMSENS 4S potentiostat, Ag/AgCl as a reference electrode, and a platinum counter electrode. All electrochemical measurements were carried out in the presence of HCO_3_^−^ to avoid any interference due to the sparging of gas mix.

### 2.8. Screening of Electrosynthesis Products

*C. saccharoperbutylacetonicum* NT-1 metabolites were screened in the catholyte and in supernatants of control cultures. Liquid samples (10 mL) were centrifuged at 7000 rpm for 10 min at 5 °C and filtered through a 0.22 μm nitrocellulose membrane. The concentration of acetate, formate, and butyrate in the catholytes was determined with high performance liquid chromatography (HPLC). Samples were filtered through 0.22 µm filter and injected in the system Vanquish Core HPLC (ThermoFisher Scientific, Hemel Hempstead, UK), equipped with Aminex HPX-87H (300 mm length × 7.8 mm diameter, 9 µm particles size) (Bio-Rad Laboratories, Watford, UK) and an UV detector operating at a wavelength of 210 nm for the organic acids and refractive index (RI) detector for ethanol. An isocratic flow of H_2_SO_4_ 0.005 M at a flow rate of 0.3 mL/min at 35 °C was used. A calibration curve with ultrapure water and known concentrations of standards was created for each analysed compound (R^2^ > 0.99). The analysis of each standard and sample was repeated twice.

### 2.9. Statistical Analysis

In order to investigate the relation between the amount of CO_2_ captured by bacteria in BESs and control cultures and the concentration of metabolites produced, we carried out a principal component analysis (PCA) [39]. Our analyses also included a clusterization of the results according to the Ward’s method and based on the Euclidean distance. All statistical tests were performed considering a significance level of 5%, using XLSTAT 2022 software. (v. 24.4.1377—Addinsoft, Paris, Ile-de-France, France).

## 3. Results and Discussion

### 3.1. Electrochemical Potential along with Biofilm Preparation

We started our experimental activity by preparing bioanodes and biocathodes from microbial cultures in rich media shifting afterwards to mineral salts formulas, and monitoring the electrodes potentials, ORP, and pH of the control cultures and of the media where the electrodes were incubated. Although connected by a 100 Ω resistor, no significant differences in the couples of carbon cloth electrodes were detected. We also report the values of electrodes potential in MFCs connected to a 1000 Ω load as the system where we measured the maximum CO_2_ reduction in the cathode compartment headspace. The results show that the ORP of cultures did not change significantly, even when using media with different composition and carbon sources (data not shown). When connected to the 1000 Ω and in the presence of HCO_3_^−^, the cathode potentials achieved −400 ± 10 mV and −422 ± 10 mV after, respectively, two and six hours, while pH values in catholyte increased to 7.1 ± 0.2. This result is consistent with previous reports indicating neutral pH values in the presence of small amounts of ethanol besides acetate when using acetogenic bacteria in MECs for CO_2_ capture [22,24]. Even though we did not investigate the presence of ethanol in catholyte, its biosynthesis cannot be excluded. As for the anodic consortium, it is interesting to notice an initial change to negative ORP in the absence of electrodes, followed by a change to positive values in control cultures inoculated in M9 medium. The pH in the anolyte (8.2 ± 0.2) was higher than in the control culture (7.8 ± 0.2). The increase in pH in both anolyte and control culture could be attributed to the excretion of metabolites such as rhamnolipids, a class of biosurfactants with alkaline properties. The ability of *Pseudomonas aeruginosa* to produce such class of metabolites from glycerol is, in fact, well known [30,31].

### 3.2. Electrochemical Behavior

We detected occasional voltage reversal in MFC_OCV during a prolonged CO_2_ bubbling at the cathode. Overall, MFC_1000 Ω kept a positive voltage along with the CO_2_ capture experiment, especially during the first hours of gas sparging. Nevertheless, when we took voltage sampling of the MFC_1000 Ω, once left in OCV, to a frequency of 0.1 s, we detected instantaneous voltage fluctuation towards negative values, from +356 to −208 mV. Electrochemical analyses are needed to investigate this behavior.

Polarization experiments give evidence of the overall electrochemical behavior of MFCs and their internal energy losses (activation, cell and mass transport losses). Figure 2 shows the results of polarization experiments carried out in MFC_1000 Ω and MFC_OCV using CO_2_ as the source of inorganic carbon. While the former showed a typical polarization curve, with an overshoot corresponding to a 100 Ω external resistor, MFC_OCV did not polarize properly, with high activation (anode) and mass losses (cathode).

The analysis of the polarization curve for MFC_1000 Ω shows a maximum power of 0.413 mW/m^2^ when connected to a 2000 Ω external resistor, and a maximum current density of 13.5 mA/m^2^ when connected to a 200 Ω resistor. For the OCV MFCs, the cell resistance was 15,000 Ω, with maximum power of 0.165 mW/m^2^ and current density of 2.06 mA/m^2^, probably formed by the Faradaic current generated by the redox species produced by the biofilm in contact with the electrode surface, as no electrons were provided by the anode [40].

The presence of electrochemical activity at the electrode colonized by *C. saccharoperbutylacetonicum* (Figure 3) revealed a reduction peak (cathodic current) at −585 mV in TSYEM with 0.25% HCO_3_^−^ (0.03 M). This value is consistent with the inactivation/reactivation midpoint potential of the hydrogenase complex observed in *Clostridium beijerinckii*, reported to be −562 mV vs. Ag/AgCl at pH 7.4 [41].

Nevertheless, the involvement of the RnF complex, a proton-translocating ferredoxin:NAD+ oxidoreductase which contributes to ATP synthesis by an H+-translocating ATPase under both autotrophic and heterotrophic growth conditions [36], and of a ferredoxin-dependent transhydrogenase as a potential entry point for extracellular electrons into the metabolism cannot be excluded [42]. This result strongly suggests the capability of *C. saccharoperbutylacetonicum* for CO_2_ biocapture in BES.

### 3.3. Inorganic Carbon Assimilation

In Table 1, we report the residual CO_2_ concentration in the headspace of both BESs and control cultures after 15 min (1st measurement) and 6 h (2nd measurement) of sparging gas mixture in the catholytes, control cultures, and sterile medium. We expressed CO_2_ concentrations as both residual CO_2_ percentage in the headspace of our systems and the calculated number of moles in the culture media of MFCs, MECs, control cultures, and abiotic controls. *C. saccharoperbutylacetonicum* NT-1 maximum CO_2_ sequestration from the gas mix (95.5%) was achieved when MFCs were connected at a 1000 Ω resistor (i.e., with no external potential applied). A reduction in the assimilation rate occurred with a prolonged exposition of the systems to the gas mix. The results suggest the ability of the consortium *S. oneidensis* MR1/*P. aeruginosa* PA1430/CO1 to sustain CO_2_ capture at the cathode. Overall, the increase in CO_2_ capture in MFC_1000 Ω and MEC was 95.5% and 76.9%, respectively, followed by MFC_OCV (70%) (Table 2). The measure of CO_2_ assimilated in abiotic controls, i.e., MEC__blank_, MFC_1000 Ω_blank, and sterile medium, revealed a reduction in CO_2_ of 55.5%, 14.9%, and 19.9%. If we consider that the reduction in CO_2_ in the MEC headspace was 79.5%, the value obtained in the MEC_blank indicates a significant contribution of abiotic processes on CO_2_ capture, in contrast to the values observed in MFC_1000 Ω_blank and MFC_1000 Ω. A possible explanation might lie in the interaction among one of the components used in the medium and the CO_2_ in the presence of a negative potential. It has been demonstrated how vitamins B can reduce CO_2_ in the absence of chemical catalysts leading to its sequestration from a gas mix [43]. From the comparison among the values of sterile medium (19.9%) with MFC_1000 Ω_blank (14.5%), and of MFC_OCV_blank (22.97%), we can estimate a contribution of about 19% decrease in CO_2_ content in gas mix due to abiotic factors. Furthermore, MFC_OCV showed a significative higher assimilation rate of CO_2_ (70%) in comparison to the control culture. As we used the same medium and same gas mix, such a difference between the control culture and MFC_OCV could be associated to the presence of biofilm growing on the cathode. It is well known how the metabolism of bacterial cells within biofilms can differ from those of planktonic cells due to nutrient and oxygen limitations, resulting in the establishment of different metabolic flux patterns [44].

### 3.4. Metabolic by-Products

The results of the analysis of formate, acetate, and 3-D-hydroxybutyrrate (3HB) are reported in Table 2. The values are expressed as average values ± standard deviation. Acetate was produced just in MEC, with an average value of 11.4 ± 1.1 mg/L*d, while the opposite happened for 3HB, whose highest amount was found in MFC_1000 Ω (2.0 ± 0.45 mg/L*d). Formate was present in all samples, with prevalence in MFC_OCV with 87.1 ± 4.4 mg/L*d. CR calculations seem to indicate that formate in MFC_OCV and in control cultures is not directly related to CO_2_ capture, as the amount found was far higher than the amount potentially produced by the sole CO_2_ assimilation. This result is confirmed by the cluster analysis, reported in Figure 4. Formate was abundant in both control cultures and MFC_OCV; therefore, in the absence of reducing equivalent inputs, it is possible that clostridia used an endogenous source of CO_2_ (for instance, from metabolic processes) to produce formate [9]. As for 3-D-hydroxybutirate, it was present in all samples except for MEC. In acetogenic bacteria, 3-D-hydroxybutyrate can be synthetized by at least two different metabolic pathways involving S)-3-hydroxybutyryl-CoA dehydrogenase (hbd1 pathway), CoA transferase/3-hydroxybutyrate dehydrogenase (CoA transferase/3 Hbdh pathway).

**Table 2 microorganisms-11-00735-t002:** Metabolytes concentration expressed as mg/L per day and carbon recovery after 6 h of CO_2_ sparging. The values are expressed as means ± SD. CR: carbon recovery expressed as a percentage.

	Formate (mg/L*d)	CR_form_ (%)	Acetate (mg/L*d)	CR_ac_ (%)	3-D-Hydroxybutyrrate (mg/L*d)	CR_but_ (%)
MFC_OCV	87.1 ± 4.4	248.99	0	0	1.7 ± 0.3	0.081
MFC_1000 Ω	31.1 ± 2.8	52.8	0	0	2.3 ± 0.4	0.066
MEC	13.9 ± 0.8	29.1	11.4 ± 1.1	18.2	0	0
Control culture	39.1 ± 1.7	193.15	0	0	0.9 ± 0.5	0.07

While both pathways involve the production of acetoacetyl CoA, the hbd1 does not entail the production of acetate nor the production of ATP, unlike the CoA transferase/3 Hbdh pathway. It is possible that the application of −0.822 mV at the cathode of MECs, by inducing the biosynthesis of acetate, might have prevented the production of 3HB: both acetate and 3HB biosynthesis starts from acetoacetyl-CoA and acetate production diverts acetyl-CoA from 3HB [45]. As for the presence of 3HB in the absence of external potential, it has been demonstrated that clostridia hbd1 is mainly involved in syngas fermentation [45]; this might explain why we observed 3HB production and no acetate in MFC_1000 Ω, MFC_OCV, and control cultures.

Figure 5 shows the results of the principal component analysis (PCA). As expected, control cultures were uncorrelated with BESs (as expected) but also with any specific metabolite. MFC1_1000 Ω was highly correlated with the CO_2_ capture values, but even in this case, the PCA does not show a significant correlation with the production of a specific metabolite. Further chemical analyses are needed to better characterize the number and amounts of metabolites in MFC_1000 Ω catholyte. MEC and MFC_OCV showed, instead, a certain correlation with, respectively, acetate and 3HB.

### 3.5. Energy Balance

We used the data collected by monitoring the power and current outputs of MFCs_1000 Ω to evaluate the energetics of CO_2_ assimilation. The calculations indicate that the amount of energy spent to capture one mole of CO_2_ by MFCs_1000 Ω was 86.4 J mol m^−2^, while the estimated energy provided by the battery during the CO_2_ capture tests in MEC was 5.7 KJ mol m^−2^. Given the results obtained in the analysis of metabolites, we can assume that a surplus of energy (in comparison to MFC_1000 Ω) was used in MECs to overcome an energy barrier, leading to the biosynthesis of acetate. In MECs, in fact, the processes occurring at the cathode are thermodynamically disadvantaged [45], thus requiring external energy input. When the energy values are reported to the net acetate amount produced and expressed as Wh mol^−1^ according to [14], the energy spent by MEC is 4.8 Wh for 1 mole of acetate. The results obtained by MEC inoculated with *C. saccharoperbutylacetonicum* NT-1 are far below what was reported by Sarakaja et al. (2019) for early operation phase (from 4 to 11 days) for MECs inoculated with *Clostridium ljunghdhalii*, with 0.2 kWh energy applied for 1 mole of acetate produced. Nevertheless, the approach based on the use of the overall energy provided to the system to calculate the energy spent to produce acetate should take into account that acetate is just part of the synthetized metabolites and the obtained values might be overestimated. For this reason, we expressed the energy efficiency as a function of CO_2_ moles captured rather than of single metabolites produced.

### 3.6. Metabolic Analysis Outcomes

Table 3 reports the AWCD values for each category of substrates and the average metabolic activity of the strains after 24 and 144 h of incubation. The prolonged incubation of *C. saccharoperbutylacetonicum* NT-1 in MFCs resulted in a change in the pattern of C-sources utilization in BioLog Ecoplates. The strains were unable to utilize an extensive range of substrates: glycogen, D-glucosamine, glucose-1-phosphate, D,L-α-glycerolphosphate, D-galacturonic acid, 2- and 4-hydroxybenzoic acid, itaconic acid, D-malic acid, L-arginine, L-phenylalanine, L-serine, L-threonine, glycyl-L-glutamic acid, phenylethylamine, and putrescine. From the analysis of the data, it is evident the higher metabolic activity of the strain from the collection and its higher capability to use carbohydrates. Interestingly, bacteria isolated from BESs after a prolonged incubation at the cathode were able to use polymers (Tween 40, tween 80, and α-cyclodextrines) and amino acids (L-asparagine), even though the expression of such phenotypes occurred at different times (24 and 144 h, respectively).

These results strongly suggest a modification of the metabolic pattern in *C. saccharoperbutylacetonicum* NT-1 as a consequence of a prolonged colonization of the cathode of MFC_1000 Ω, i.e., in the presence of reducing equivalents provided by the anode, supporting growth in a minimal medium with inorganic carbon as sole source of carbon.

To the best of our knowledge, this is the first time that the utilization of MFCs has been attempted for direct CO_2_ capture and its conversion into organic compounds by using pre-colonized biocathodes and bioanodes. MFC_1000 Ω achieved a CO_2_ capture of 95.5% with *P. aeruginosa P01/S. oneidensis*-MR1 anodic consortium supplying electrons to *C. saccharoperbutylacetonicum* NT-1 biocathode, whose redox potential achieved −422 ± 10 mV vs. Ag/AgCl, able to provide a sufficiently low potential to feed into the cellular NADH pool (about −280 mV required) [42]. It has been reported that redox mediators with a lower redox potential affect the metabolism of *Clostridium autoethanogenum* cultured in the presence of applied potential and using fructose as a source of carbon [36]. Instant cell voltage reversal in MFC_1000 Ω might have played a role in the overall CO_2_ conversion process, although further studies are needed to understand the mechanisms that occurred at the cathode of MFCs. Although MFC_1000 Ω was the most efficient system in capturing CO_2_, the analysis of formate, acetate, and 3HB showed the prevalence of formate (7.78 mg/L, 52.8% CR), whose biosynthesis requires a low redox potential and energy efficiency in comparison to other biosynthetic processes [46]. Formate was also prevalent in MFC_OCV and in control cultures, where we detected the highest production (21.8 mg-L and 39.1 mg/L). However, according to our data, such production cannot be entirely ascribed to the assimilation of CO_2_ from gas mix, but very likely of CO_2_ produced by the microbial cells themselves. No acetate was produced, but at least traces of 3-D-hydroxybutirate (0.57 mg/L, 0.06% CR in MFC_1000 Ω, 0.41 mg/L, 0.08% CR and 0.22 mg/L, 0.08 CR in control cultures) were found after 6 h of gas sparging. As for the CO_2_ capture, MFC_OCV achieved about 70%, while just the 22% of CO_2_ was removed by the gas mix after 15 min.

Further investigations about the synthesis of other metabolites should be carried out. A lower CO_2_ capture rate was observed in MEC (76.9%), with concurrent production of acetate and formate in MEC of, respectively, 2.4 mg/L and 3.48 mg/L (with a CR of 18.2% and 29.1%, respectively) within 6 h of operation using a gas mix with 2% of CO_2_, much lower than the concentration used in previous studies with clostridia in MECs [22,24,27,28]. The use of a pre-formed biocathode and bioanode reduced the start-up period, with metabolites production (acetate included) produced since the first few hours of operation, unlike what was reported by other authors who tested clostridia in their BESs [16,46]. Previous reports showed production of acetate by applying −0.8 V at the cathode, with a start-up period between 2 and 10 days, with, respectively, an inoculum of acetogenic bacteria and *Clostridium ljundghali* [17,24,28].

From the analysis of the results, abiotic factors such as medium composition, temperature, and the cathodes’ redox potential can affect the amount of CO_2_ available for microbial metabolism, as suggested by the 55.5% of carbon dioxide decrease in MEC_blank.

Under an energy utilization perspective, the use of pre-constituted bioelectrodes led to a reduction in the overall energy required to produce acetate 4.8 Wh mol^−1^ in MEC similarly to what was reported by Bajracharya et al. (2015) [28] and Noori et al. (2021) [17]. Undoubtedly, the preparation and acclimatization of the biocathode prior to their use in BESs overcome some limitations in MECs performance, linked to factors such as pH and electrostatic repulsion among cathode and bacterial surfaces (both with negative charges), negatively affecting biofilm formation and metabolism of acetogenic bacteria at the cathode [47], besides taking to a reduction in the overall energy demand for both direct CO_2_ capture and acetate biosynthesis. If we consider that one of the factors that might limit the future development of MET is the energy demand for “driving” the processes at the biocathodes, our results, if further confirmed, might open new possibilities for future practical exploitation of BESs [48].

By preparing biocathodes in controlled conditions, shifting from rich to minimal media and from organic to inorganic carbon source, we achieved a very high CO_2_ capture rate in two-chamber microbial fuel cells. The analysis of the metabolic pattern through the ECOLOG multiwell assay revealed a change in the ability to use carbon substrates in *C. saccharoperbutylacetonicum* NT-1 after a prolonged operation in MFC_1000 Ω, demonstrated by the utilization of polymers and amino acids. Such change in the utilization of carbon sources might be a response to the limitation of carbon sources available for the growth of *Clostridium* in the minimal medium at the cathode and in the presence of CO_2_. It is well known how environmental conditions such as carbon sources and nutrients availability, the chemical physical condition realized in the growth medium (ionic concentration, pH, etc.), as well as electrochemical potential and temperature can exercise a selective pressure on microorganisms [49]. Therefore, the growth of a microorganism for several thousand generations at the cathode of a MEC, without organic compounds as a source of carbon and exposed at a well-defined range of electrochemical potentials, might have induced metabolic adaptation and a modification in the pattern of carbon source that can be used to sustain the microbial growth.

## 4. Conclusions

In the present research, we aimed at maximizing the efficiency of CO_2_ capture at the cathodes of our BESs by minimizing any potential factors preventing or hampering the formation of a cathodic biofilm in MECs. Therefore, we prepared biocathodes under conditions facilitating biofilm formation and used culture media to support a gradual metabolic adaptation in *Clostridium saccharoperbutylacetonicum* NT-1 towards the utilization of inorganic carbon. The overall effect was the constitution of an electroactive biofilm, with an activated WL pathway and ready to use CO_2_ at the cathode of BESs, with high capture yield in MFCs and significant reduction in the overall start-up phase. We demonstrated the possibility to set up MFCs able to capture CO_2_, with prevalent biosynthesis of formate, with low amount of 3HB, while acetate and formate were found in MEC. Nevertheless, the biosynthesis of other metabolites, under the different operational conditions realized in this study, should be investigated. We also show for the first time that *C. saccharoperbutylacetonicum* NT-1, a strain used in conventional industrial fermentation processes, can also be used in BESs for the electrosynthesis of organic compounds from CO_2_. The successful use of glycerol at the anode suggests other possible applications, such as the biosynthesis of organic compounds of industrial interest (e.g., rhamnolipids) besides by-products of inorganic carbon metabolism. Furthermore, the analysis of *C. saccharoperbutylacetonicum* NT-1 metabolic patterns allowed us to detect changes in the ability of bacteria to use different carbon sources, providing understanding of the physiology of electroactive microorganisms and its future exploitation. If successfully delivered, systems not requiring external energy input may simplify the design of BESs for CO_2_ capture and concurrent electrosynthesis of platform chemicals, fostering the set up of scaled-up systems and reducing the time required to achieve a proper industrial development for practical applications.

## Figures and Tables

**Figure 1 microorganisms-11-00735-f001:**
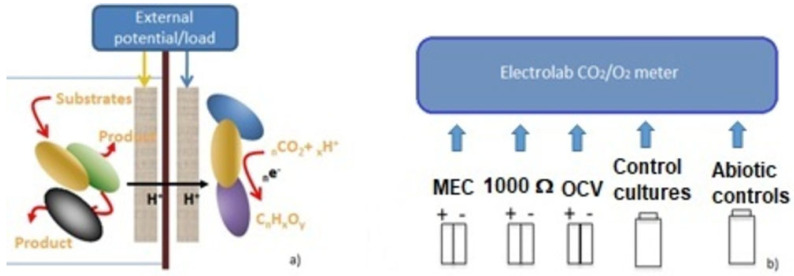
Experimental set-up. (**a**) Schematic representation of the processes occurring at the electrode of MFCs and MECs used in our research. (**b**) MFCs and MECs and controls used for CO_2_ capture testing.

**Figure 2 microorganisms-11-00735-f002:**
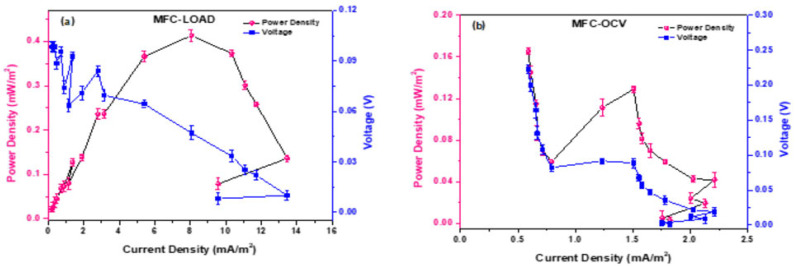
Power and polarization curves of (**a**) MFC_1000 Ω and (**b**) MFC_OCV. The arrows indicate anodic (orange arrows) and cathodic energy losses (blue arrows).

**Figure 3 microorganisms-11-00735-f003:**
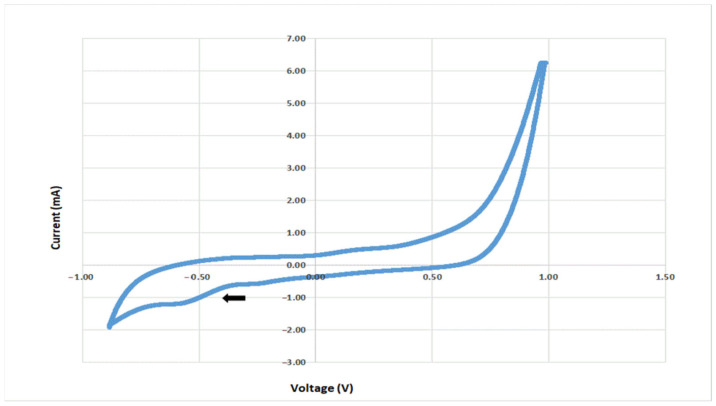
CV (1 mV/s) of *C. saccharoperbutylacetonicum* biofilm in TSYE with HCO_3_^−^ 0.25% (0.03 M). The arrow indicates a reduction peak at −585 mV vs. Ag/AgCl.

**Figure 4 microorganisms-11-00735-f004:**
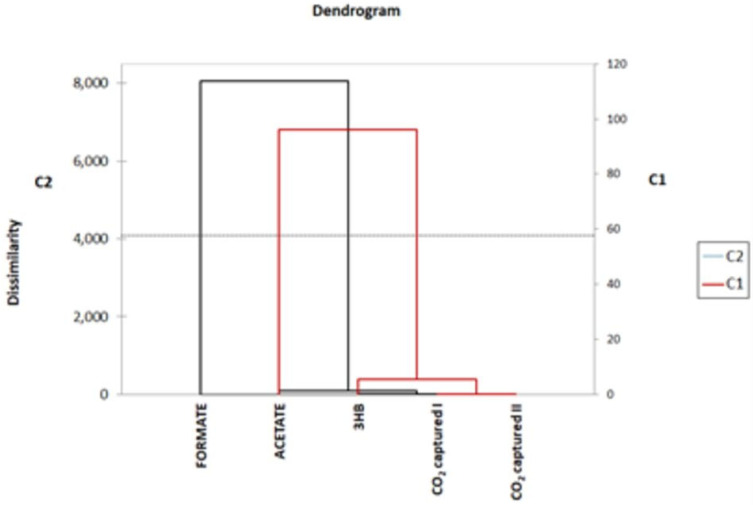
Cluster analysis based on dissimilarity of formate, acetate, 3HB, and CO_2_ capture values. Formate forms a separate cluster from acetate, 3HB, and CO_2_ captured I.

**Figure 5 microorganisms-11-00735-f005:**
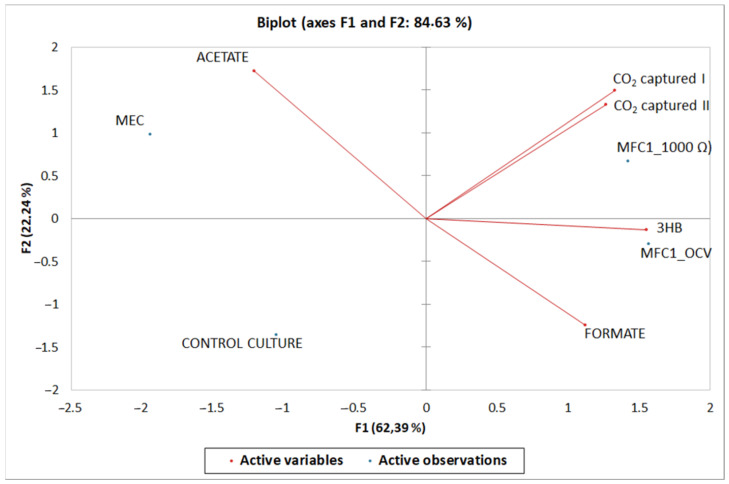
PCA diagram (5% significance level). 3HB = 3-D-Hydroxybutirate; CO_2_ captured I and CO_2_ captured II are the mol of CO_2_ captured after 15 min and 6 h of gas sparging.

**Table 1 microorganisms-11-00735-t001:** CO_2_ removal in MFC_+1.5 V: MFCs with 1.5 V applied; MFC_1000 Ω: MFC kept at 1000 Ω; MFC (blank): MFC fed with sterile medium. * CO_2_ removal after 15 min (° 1 measurement) and ’ CO_2_ removal after 6 h of gas mix sparging in the catholyte (° 2 measurement).

Sample	Residual CO_2_ in the Headspace (%)	Residual CO_2_ in the Headspace (%)	Residual Concentration in the Medium (M)	Residual Concentration in the Medium (M)	nCO_2_ Captured (mol)	nCO_2_ Captured (mol)	% CO_2_ Captured (mol)	% CO_2_ Captured (mol)
	1st Measurement *	2nd Measurement ’	1st Measurement *	2nd Measurement ’	1st Measurement *	2nd Measurement ’	1st Measurement *	2nd Measurement ’
MEC	0.46	1.34	0.078	0.227	0.261	0.112	76.99	32.98
MFC_OCV	0.595	1.38	0.101	0.234	0.238	0.105	70.24	30.98
MFC_1000 Ω)	0.09	1.43	0.015	0.242	0.324	0.097	95.50	28.47
MEC (blank)	0.89	1.45	0.151	0.246	0.188	0.093	55.48	27.47
MFC(blank)_1000 Ω	1.7	1.58	0.288	0.268	0.051	0.071	14.97	20.97
MFC(blank)_OCV	1.54	1.57	0.261	0.266	0.078	0.073	22.97	21.47
Control culture	1.55	1.65	0.263	0.280	0.076	0.059	22.47	17.47
Abiotic control	1.6	1.7	0.271	0.288	0.068	0.051	19.97	14.97

**Table 3 microorganisms-11-00735-t003:** AWCD values calculated for the four classes of carbon sources. The average values are calculated based on the reading of all 96 wells.

	AwCD_BES_24 h	AwCD_Coll_24 h	AwCD_BES_144 h	AwCD_Coll_144 h
Carbohydrates	0.009	0.027	0.0794	0.637
Polymers	0.019	0	0	0
Ammines	0	0	0	0
Amminoacids	0	0	0.085	0
Phenolic Compounds	0	0	0	0
Average	0.005	0.009	0.101	0.219

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
