# Peer review of "Inorganic Carbon Assimilation and Electrosynthesis of Platform Chemicals in Bioelectrochemical Systems (BESs) Inoculated with Clostridium saccharoperbutylacetonicum N1-H4"

_microorganisms, 2023, doi:10.3390/microorganisms11030735_

Round 1
Author Response
We gratefully thank the reviewer for their valuable comments.
The authors investigated inorganic carbon assimilation and electrosynthesis of platform chemicals in Bioelectrochemical Systems. However, present version is not allowed for considering publication. Several uncertain information needs to be clarified. The manuscript might be published to journal if the authors would be able to modify manuscript in accordance with several comments below.
Thank you very much for your comment. We carried out a deep revision of our manuscript by adding further details about the methodology applied, the results and their significance and our conclusions. Further we moved some information in a dedicated file (supporting information).
- It was hard to understand methodology. Have the authors conducted MFC and MEC experiments? But only MFC data was shown in the result. I don't see the role of MEC in this study. Please make operational condition and methodology much clear.
Thank you very much for your comments. Our experimental scheme included both MFCs and MECs in order to investigate the response of C.saccharoperbutylacetonicum NT1 metabolism to different operational conditions and identify the configuration able to give the highest CO2 assimilation rate and the energy spent by the systems to carry out the conversion of inorganic carbon to organic compounds. In order to simplify the layout of MECs to be potentially used in CO2 biocapture, we connected 1.5V battery to our system, applying -750 mV potential (vs SHE) at the cathode and verify its effect on the ability of C.saccharoperbutylacetonicum to assimilate CO2. The choice was to simplify MECs layout, possibly avoiding the connection to a potentiostat. [see Giddings CG, Nevin KP, Woodward T, Lovley DR, Butler CS. Simplifying microbial electrosynthesis reactor design. Front Microbiol. 2015 May 15;6:468. doi: 10.3389/fmicb.2015.00468. PMID: 26029199; PMCID: PMC4432714; Noori M.T., Mohan S.V., Min B. Microbial electrosynthesis of multi-carbon volatile fatty acids under the influence of different imposed potentials, Sustainable Energy Technologies and Assessments, Volume 45, 2021, 101118, ISSN 2213-1388, https://doi.org/10.1016/j.seta.2021.101118].
- When did the authors conduct experiment for analyzing CV? In the turnover or unturnover condition? And why there is only one CV curve?
CV was carried out on the cathode electrode before being used in MFCs, so soon after the incubation period and having the bacteria use HCO3- as source of carbon and energy, during the turnover condition. We didn’t perform CVs of the cathodes in the MFCs and MECs during their operation, but we decided to add this CV as, in any case, it might offer the possibility of (limited) electrochemical interactions among bacteria and the conductive electrode, made of carbon cloth. [Funari R., ShenA.Q. Detection and Characterization of Bacterial Biofilms and Biofilm-Based Sensors. ACS Sens. 2022, 7, 2, 347–357. https://doi.org/10.1021/acssensors.1c02722]
- The serial number in Figure 4 is not uniformly marked and arranged in an orderly manner.
- REPLY: we revised Figure 4
- How many days and circles of experiments have the authors done in this study? 5) How did the authors monitor the current in the OCV state in a MFCs (Fig. 2c)?
The data reported in this paper refers to three cycles of experiments (without considering some preliminary experimental activities), with bioelectrochemical systems and control cultures in double replicas. The duration of each cycle was about three weeks, considering also the preparation of pre-colonized bioanodes and biocathodes. We added the following text at the section 2.3:
All MECs, MFCs and controls were set-up in double replicas for three cycles of experimental activity, performed at 20°C.
Fig 2 reports the outcome of polarization experiments we performed to MFCs. The diagram reports both power and current production as consequence of the application of external loads of different values (from 120kΩ to 100Ω) for 5 minutes. A fuel test equipment (ARBIN) measured voltage variations while current and power densities were calculated according Ohm’s law. We added further explanations in the section 3.2.
While revising the paper, we also realized there was a problem with the data we previously used to produce the power and polarization curves of our MEC. We deleted it.
- The discussion Section is too short and all data should be discussed in the manuscript.
We improved the discussion of our data pointing out the significance of our results in comparison to previous studies.
7) It is recommended to add a graphic abstract to the manuscript to help readers better understand this study.
We prepared a graphical abstract according to reviewer’s recommendation
Reviewer 2 Report
Dear Authors,
The Authors show that MFCs have the potential to contribute to CO2 capture from gas mixtures, enhancing the yield of the metabolic process leading from an inorganic source of carbon to platform chemicals, and combining it with the synthesis of bio surfactants at the anode without any external energy. The monitoring of microbial metabolism, along with the utilization of Clostridium in the MFCs, let us detect the change in the ability of bacteria to use different carbon source and opens to a better understanding of microbial physiology in presence of electro genesis and its future exploitation.
The description of the work is acceptable. Overall impression is that this manuscript can be recommended for publication after MAJOR revision in Microorganisms especially considering the scope and topics of this journal. However, I would like to point out to several details:
- In my opinion Section 1.1. and 1.2. should be connected as one Section – Introduction. Correct this.
- Please note in the Instruction of the Microorganisms for Introduction and state the objectives of the work and provide an adequate background, avoiding a detailed literature survey or a summary of the results. Correct this.
- It is not clear what novelty in paper worth to publish is? Correct this.
- The data of methods that include in this paper is dependent on the matrix. That effect is very important in the real samples but the authors did not explain the effect of the matrix. Correct and explain this.
- The authors should avoid vertical rules in the tables. All tables should be in the same format. Correct this.
- In the conclusions, in addition to summarizing the actions taken and results, please strengthen the explanation of their significance. It is recommended to use quantitative reasoning comparing with appropriate benchmarks, especially those stemming from previous work. This should be corrected.
- English language should be corrected by a professional lector. A proof reading by a native English speaker should be conducted to improve both language and organization quality.
I wish a lot of success to the authors in making this manuscript much better.
With kind regards!
Reviewer
Author Response
We thank the reviewer for their useful comments. We have carefully considered the suggestions, as follows.
The description of the work is acceptable. Overall impression is that this manuscript can be recommended for publication after MAJOR revision in Microorganisms especially considering the scope and topics of this journal. However, I would like to point out to several details:
In my opinion Section 1.1. and 1.2. should be connected as one Section – Introduction
We joined the two sections as advised
Please note in the Instruction of the Microorganisms for Introduction and state the objectives of the work and provide an adequate background, avoiding a detailed literature survey or a summary of the results. Correct this.
We revised the whole introduction clearly stating the objectives of the research while widening the discussion of its background
We added a supporting information file where we provided further details about the methodological approach we used and moved some details about the calculations we did.
It is not clear what novelty in paper worth to publish is? Correct this.
In this paper, we demonstrate the effect of biocathode and bioanode preparation before being used in BESs on the reduction of start-up phase of our system. We also demonstrate the possibility of simplifying the layout of BESs for CO2 capture by using a simple 1.5V battery or even avoiding its utilization by developing an anodic consortium formed by highly electroactive microorganisms. Such aspect might take BESs closer to an industrial application. We also show for the first time that C.saccharoperbutylacetonicum (a strain used in conventional fermentation processes for production of acetone and butanol) is used in bioelectrochemical systems (MFCs AND MECs) for the electrosynthesis of organic compounds from CO2.
We added the following text to the Introduction section:
Within this framework, we assessed the efficiency of a fermentative microorganism such as Clostridium saccharoperbutylacetonicum NT-1, used in industrial processes for the production of bio-butanol and acetone [24], to convert CO2 into acetate and D-3 hydroxybutyrrate, in presence and in absence of an external source of electrochemical potential.We used a consortium Pseudomonas aeruginosa PA1430/CO1 /Shewanella oneidensis MR_1 as bioanode and investigated their ability to sustain CO2 assimilation from a gas mix in C. saccharoperbutylacetonicum NT-1 biocathodes, previously established in controlled growth conditions. To the best of our knowledge, this is the first time that C.saccharoperbutylacetonicum NT-1 and potential modifications in its metabolism in terms of utilization of carbon sources has been investigated in BESs for CO2 biocapture, and an approach envisaging the utilization of both MFCs, MECs and pre-constituted bioelectrodes has been used.
The data of methods that include in this paper is dependent on the matrix. That effect is very important in the real samples but the authors did not explain the effect of the matrix. Correct and explain this.
We agree with the reviewer that the culture medium affects the overall microbial metabolism and, also, the production of energy in bioelectrochemical systems. Nevertheless, electrosynthesis is a process requiring well-controlled experimental conditions, so it is mnot expected to happen in wastewater or other complex substrates. The use of media containing carbon sources other than CO2 may result in the establishment of a fermentative metabolism in C.saccharoperbutylacetonicum that would compete with the CO2 capture. Therefore, it is important to use minimal media with no other source of carbon or energy.
To account for this issue, we included the following explanation to the introduction
The performance of MET can vary according to the operational conditions, the microbial species, and the substrates used: for example, organic substrates in the catholyte might reduce or even prevent CO2 assimilation, fostering the establishment of heterothrophic instead of an autothrophic or mixothrophic metabolism. Therefore, the set-up of MET devices requires adequate preliminary studies [17][21].
The authors should avoid vertical rules in the tables. All tables should be in the same format. Correct this.
We revised the tables format and deleted the vertical rules.
In the conclusions, in addition to summarizing the actions taken and results, please strengthen the explanation of their significance. It is recommended to use quantitative reasoning comparing with appropriate benchmarks, especially those stemming from previous work. This should be corrected.
We fully revised the conclusions as follows:
In the present research, we aimed at maximizing the efficiency of CO2 biocapture at the cathodes of our BES by minimizing any potential factors preventing or hampering the formation of a cathodic biofilm in MECs. Therefore, we prepared biocathodes under conditions to facilitate biofilm formation, and used culture media with different to supporting a gradual metabolic adaptation in Clostridium saccharoperbutylacetonicum NT-1 towards the utilization of inorganic carbon. The overall effect was the constitution of an electroactive biofilm, with an activated WL pathway and ready to use CO2 at the cathode of BESs, with high capture yield in MFCs and significant reduction of the overall start-up phase. We demonstrated the possibility to set-up MFC able to biocapture CO2, with prevalent biosynthesis of acetate and probably other metabolites, under the different operational conditions realized in this study. We also show for the first time that C.saccharoperbutylacetonicum NT-1, a strain used in conventional industrial fermentation processes can also be used in BESs for the electrosynthesis of organic compounds from CO2. The successful use of glycerol at the anode suggests other possible applications, such as the biosynthesis of of organic compounds of industrial interest (e.g. rhamnolipids) besides byproducts of inorganic carbon metabolism. Furthermore, the analysis of C.saccharoperbutylacetonicum NT-1 metabolic patterns allowed us to detect changes in the ability of bacteria to use different carbon sources, providing understanding of the physiology of electroactive microorganisms and its future exploitation. If successfully delivered, systems not requiring external energy input may simplify the design of BESs for CO2 capture and concurrent electrosynthesis of platform chemicals, fostering the set-up of scaled-up systems and reducing the time required to achieve a proper industrial development for practical applications
English language should be corrected by a professional lector. A proof reading by a native English speaker should be conducted to improve both language and organization quality.
A proficient speaker proofread the manuscript, and we have moved parts to the supplementary materials section, resulting in a more streamlined and better organized manuscript.
Reviewer 3 Report
Interesting article about MFCs (Microbiological Fuel Cells).
I propose the following corrections and additions:
1. Abstract - line 28- please add the expanation of MFCs (Microbiological Fuel Cells)
2. Introduction - lines72-78- please specify the aim of Your research
3. Section 2.3. - line 147 - It should be "Figure 1"
4. Section 3.2. - line 307 - It should be "Figure 2"
5. Section 3.2. - line 322 - It should be "Figure 3"
6. Conclusions - please reword this section, when You specify the aim of research
Author Response
We gratefully thank the reviewer for contributing to improve our manuscript through their comments.
I propose the following corrections and additions:
- Abstract - line 28- please add the explanation of MFCs (Microbial Fuel Cells)
- We added the explanation of MFCs acronym in the text
- Introduction - lines72-78- please specify the aim of Your research
- Thank you for pointing out this oversight. We added a description of the scope of the research, as shown below:
Within this framework, we assessed the efficiency of a fermentative microorganism such as Clostridium saccharoperbutylacetonicum NT-1, used in industrial processes for the production of bio-butanol and acetone [243], to convert CO2 into acetate and D-3 hydroxybutyrrate, in presence and in absence of an external source of electrochemical potential.We used a consortium Pseudomonas aeruginosa PA1430/CO1 /Shewanella oneidensis MR_1 as bioanode and investigated their ability to sustain CO2 assimilation from a gas mix in C. saccharoperbutylacetonicum NT-1 biocathodes, previously established in controlled growth conditions. To the best of our knowledge, this is the first time that C.saccharoperbutylacetonicum NT-1 and potential modifications in its metabolism in terms of utilization of carbon sources has been investigated in BESs for CO2 biocapture, and an approach envisaging the utilization of both MFCs, MECs and pre-constituted bioelectrodes has been used.
- Section 2.3. - line 147 - It should be "Figure 1"
- We changed the text from Figure 2 to figure 1
- Section 3.2. - line 307 - It should be "Figure 2"
- We amended the text according to the reviewer’s remark
- Section 3.2. - line 322 - It should be "Figure 3"
- We revised the text
- Conclusions - please reword this section, when You specify the aim of research
- We modified the conclusions as follows:
In the present research, we aimed at maximizing the efficiency of CO2 biocapture at the cathodes of our BES by minimizing any potential factors preventing or hampering the formation of a cathodic biofilm in MECs. Therefore, we prepared biocathodes under conditions to facilitate biofilm formation, and used culture media with different to supporting a gradual metabolic adaptation in Clostridium saccharoperbutylacetonicum NT-1 towards the utilization of inorganic carbon. The overall effect was the constitution of an electroactive biofilm, with an activated WL pathway and ready to use CO2 at the cathode of BESs, with high capture yield in MFCs and significant reduction of the overall start-up phase. We demonstrated the possibility to set-up MFC able to biocapture CO2, with prevalent biosynthesis of acetate and probably other metabolites, under the different operational conditions realized in this study. We also show for the first time that C.saccharoperbutylacetonicum NT-1, a strain used in conventional industrial fermentation processes can also be used in BESs for the electrosynthesis of organic compounds from CO2. The successful use of glycerol at the anode suggests other possible applications, such as the biosynthesis of of organic compounds of industrial interest (e.g. rhamnolipids) besides byproducts of inorganic carbon metabolism. Furthermore, the analysis of C.saccharoperbutylacetonicum NT-1 metabolic patterns allowed us to detect changes in the ability of bacteria to use different carbon sources, providing understanding of the physiology of electroactive microorganisms and its future exploitation. If successfully delivered, systems not requiring external energy input may simplify the design of BESs for CO2 capture and concurrent electrosynthesis of platform chemicals, fostering the set-up of scaled-up systems and reducing the time required to achieve a proper industrial development for practical applications
Round 2
Reviewer 1 Report
No comments
Author Response
We gratefully thank the reviewer for their significant contribution to the improvement of our manuscript.
Reviewer 2 Report
Dear Authors,
Interesting results are well presented. The description of the work is acceptable. The length of the manuscript is appropriate. Discussion and conclusion is detailed. In my opinion this manuscript can be PUBLISH in Microorganisms especially considering the scope and topics of this journal. The authors correct all suggestions that reviewers gave about article.
I wish a lot of success to the authors.
Regards!
Reviewer
Author Response
We gratefully thank the reviewer for helping us improving our manuscript and for their positive comment.